# Functionalized Dendrimer Platforms as a New Forefront Arsenal Targeting SARS-CoV-2: An Opportunity

**DOI:** 10.3390/pharmaceutics13091513

**Published:** 2021-09-18

**Authors:** Serge Mignani, Xiangyang Shi, Andrii Karpus, Giovanni Lentini, Jean-Pierre Majoral

**Affiliations:** 1Laboratoire de Chimie et de Biochimie Pharmacologiques et Toxicologique, Université Paris Descartes, PRES Sorbonne Paris Cité, CNRS UMR 860, 75006 Paris, France; 2CQM—Centro de Química da Madeira, MMRG, Campus da Penteada, Universidade da Madeira, 9020-105 Funchal, Portugal; 3College of Chemistry, Chemical Engineering and Biotechnology, Donghua University, Shanghai 201620, China; 4Laboratoire de Chimie de Coordination du CNRS, 205 Route de Narbonne, CEDEX 4, 31077 Toulouse, France; andrii.karpus@lcc-toulouse.fr; 5Université Toulouse 118 Route de Narbonne, CEDEX 4, 31077 Toulouse, France; 6Dipartimento di Farmacia—Scienze del Farmaco, Università degli Studi di Bari Aldo Moro, 70125 Bari, Italy; giovanni.lentini@unida.it

**Keywords:** COVID-19 pandemic, SARS-CoV-2, nanotechnology, dendrimers, repurposing strategy

## Abstract

The novel human coronavirus SARS-CoV-2 (severe acute respiratory syndrome coronavirus 2) has caused a pandemic. There are currently several marketed vaccines and many in clinical trials targeting SARS-CoV-2. Another strategy is to repurpose approved drugs to decrease the burden of the COVID-19 (official name for the coronavirus disease) pandemic. as the FDA (U.S. Food and Drug Administration) approved antiviral drugs and anti-inflammatory drugs to arrest the cytokine storm, inducing the production of pro-inflammatory cytokines. Another view to solve these unprecedented challenges is to analyze the diverse nanotechnological approaches which are able to improve the COVID-19 pandemic. In this original minireview, as promising candidates we analyze the opportunity to develop biocompatible dendrimers as drugs themselves or as nanocarriers against COVID-19 disease. From the standpoint of COVID-19, we suggest developing dendrimers as shields against COVID-19 infection based on their capacity to be incorporated in several environments outside the patients and as important means to stop transmission of SARS-CoV-2.

## 1. Introduction

Pandemics have affected civilizations throughout human history, with the first known epidemic occurring during the Peloponnesian War in 430 BC. The current outbreak of a new SARS-CoV-2 causes a disease known as COVID-19. In a rapid historical way, in late December 2019 several health facilities in Wuhan, located in the Hubei province of China, reported groups of patients with pneumonia of unknown cause but similar to patients with SARS and MERS, with symptoms including fever, cough and chest discomfort, and in severe cases dyspnea and bilateral lung infiltration [1,2]. According to hospital reports, most cases were epidemiologically linked to the Huanan Seafood Wholesale Market. This market sells not only seafood but also live animals, including poultry and wildlife [3]. On December 31, the World Health Organization (WHO) was informed of an outbreak of pneumonia of unidentified cause in the Wuhan region following public information [4]. Due to the rapid, global spread of the catastrophic COVID-19 pandemic, many health systems have been overwhelmed by the disease’s rapid emergence and high levels of new cases, revealing the global community’s response. Virus transmission can occur through physical contact, encompassing coughing, sneezing, speaking and singing [5], and can produce both mild and more severe symptoms, such as the loss of smell and taste and critical lung respiratory failure, including acute pneumonia, multiorgan failure, and an uncontrolled inflammatory response (named ‘cytokine storm’), unfortunately resulting in death in the most serious cases [6]. Importantly, recent reports have highlighted that many infected people show no symptoms. Consequently, it is quite impossible to precisely evaluate the rate of infection in the general population. Currently, an important issue is the apparition of mutations in various regions of the world [7]. From January 2020 onwards, the COVID-19 global health pandemic has led to a frenetic search for potential therapies and preventive measures. Consequently, a tsunami of papers and reviews have been published, which continues today, enabling better comprehension of all aspects of COVID-19 infection; these studies include details on many potential virus-based and host-based targets against viral entry, transcription and translation, genome synthesis and assembly within the replication cycle, as well as clinical lessons learned and novel drugs [8], vaccines and therapeutic molecules [9]. Additionally, new biological agents have been developed, for instance those aimed at controlling the cytokine storm (vide infra).

At the forefront of approaches for tackling the COVID-19 pandemic are some concomitantly developed strategies [9] worth noting, particularly drug repurposing [10,11] and the development of vaccines targeting specific strains of the virus [12]. A tidal wave of repurposing candidate trials rapidly emerged, however no effective drugs targeting SARS-CoV-2 are currently on the market [13,14]. This repurposing strategy involves the selection of existing approved drugs, with various targets and mechanisms of action, for treatment of COVID-19, either alone or in combination, and includes many potentially targetable steps of the coronavirus life cycle [15]. Importantly, there is always a risk when combining numerous substances. An interesting analysis was performed by McKeigue et al. showing that severe COVID-19 is strongly associated with inappropriate polypharmacy [16]. On 11 May 2020, the novel use of already established drugs, such as hydroxychloroquine and chloroquine, was approved in the treatment of COVID-19 infection through the EUA process, despite very limited and controversial clinical evidence [17]. On 15 June 2020, the FDA revoked its EUA for hydroxychloroquine due to a lack of efficacy. Recently, a critical analysis was conducted by Boener [18]. Nearly 60% of the Phase III clinical trials used hydroxychloroquine or chloroquine, with an estimated cost over USD 6 billion in the past year. However, the drug repurposing strategy against the COVID-19 disease remains a very interesting approach and is based on previous successes in the clinical setting, for instance those of Viagra and Thalidomide [19]. Several drug categories have been selected and screened for use in COVID-19 therapy, including: (1) antiviral drugs [15]; (2) anti-malaria drugs [20]; (3) cardiovascular drugs [15]; (4) various traditional Chinese medicines [15,21]; (5) natural plant products [22]; (6) psychotropic drugs [23]; (7) inhibitors of SARS-CoV-2S-protein from in silico analysis [24,25] and virtual screening [26,27,28]; (8) drugs blocking SAR-CoV-2 endocytosis [29]; (9) antiseptic quaternary ammonium drugs with broad-spectrum antiviral activities [30]; (10) anticancer drugs [31,32]; (11) anti-HCV drugs and nucleotide inhibitors [33]; (12) adamantane derivatives [34]; (13) host transcriptome-guided drug repurposing candidates [35]; (14) drugs for leprosy treatment [36]; (15) lipid lowering agents [20]; (16) rheumatoid arthritis drugs [20]; and (17) SiRNAs therapeutics [37]. It is also important to note that Pfizer is currently developing novel protease inhibitors against COVID-19, including PF-07304814 and PF-07321332 (oral administration) that are in Phase I clinical trials [38].

Importantly, the development of anti-inflammatory drugs as complementary therapeutics for treating COVID-19 is important with regards to the development of the uncontrolled “cytokine storm” that leads to lethal lung injury. In this direction, the combination of thalidomide, acting as an immunomodulator and anti-inflammatory agent, with low-dose glucocorticoids was reported in the treatment of a COVID-19 patient with severe COVID-19 pneumonia; cytokine surge inhibition, immune function regulation, reduced oxygen consumption and relief of digestive symptoms were all observed [39]. Several additional clinical trials have been registered [40], and similarly, antibodies against the interleukin 6 (IL-6) receptor [41,42] and IL-6 itself are under development for the treatment of the acute respiratory distress syndrome (ARDS) suffered by some COVID-19 patients [42]. Another direction is the development of neutralizing monoclonal antibodies such as 47D11 humAb that binds to the S protein RBD and can neutralize SARSCoV-2 infection [43].

From a cursory glance, over the last decade, nanotechnology has opened up an extensive realm of research and application. Within this field, the aim of nanomedicine, which uses various powerful nanotechnologies and specific nano-objects, is to optimally solve specific clinical problems, combat diseases and find solutions to public health problems. Interactions of these nanodevices with biological molecules at a nanoscale are based on the characteristics of these nanoparticles, such as the volume/surface ratio, nature of the groups on the surface, shape and surface charge [44,45]. Nanomedicine is poised to revolutionize medicine through the development of more precise diagnostic and therapeutic tools. The field of nanomedicine encompasses many therapeutic features and disciplines [46]. The aim is to develop innovative applications in the health field by exploiting the intrinsic physical, chemical and biological properties of these versatile nanoscale materials for precision medicine [45,47]. Precision nanoparticles, such as, for instance, PEGylated and non-PEGylated liposomes, micelles, nanocrystals, polymer-drug conjugates, polymer-protein conjugates, polyplexes, degradable nanogels and dendrimers, have been designed and developed for the application of nanotechnology in nanomedicine [48].

Dendrimers are synthetic homogeneous nano-sized symmetric macromolecules with nearly monodisperse structure. The design of dendritic macromolecules is a relatively new field in nanomedicine developed by Tomalia et al. [49]. For instance, drug delivery [50] was pioneered in 1978 by Vogtle and colleagues [51]. Then, Tomalia et al. named this new class of versatile NPs “dendrimers”, formed from the two Greek words “dendros”, meaning “tree” or “branch”, and “meros”, meaning “part” [52].

As shown in Figure 1, related to the well-known polyamidoamine (PAMAM) dendrimers, these hyperbranched macromolecules contain symmetric branching elements, which are built around a central core where dendrimer growth begins and that allow internal cavities [53,54]. Formed of repetitive units geometrically organized under radial layers, dendrimers are often classified based on their generation number (e.g., G0-Gn). Importantly, dendrimers have a distinct ramified and tailored architecture that allows for a diverse set of functionalized moieties to be introduced on their surfaces, thereby enabling fine-tuning of their physicochemical and/or biological properties [55,56]. To date, there are over 100 families of dendrimers [57]. Importantly, biologically targeted active compounds, such as small molecules, macromolecules, peptides and metal nanoparticles, can be encapsulated inside the void spaces of dendrimers, whereas small molecules, macromolecules, targeting peptides, antibodies and nucleic acids can be conjugated or complexed with the end surface groups [58,59,60]. The main objective of dendrimer application is to improve the therapeutic outcomes of the loaded drugs, such as their pharmacokinetic (PK)/pharmacodynamic (PD) profiles [61]. These nanodevices can be considered “Trojan horses” for targeting specific organs, tissues and sites of inflammation in COVID-19 patients (vide infra). Low polydispersity and biocompatibility are the two main challenges to be considered for the use of dendrimers in nanomedicine, both as nanocarriers and as drugs (active per se) [62,63,64]. Many studies have highlighted dendrimers as nanocarriers [65], for instance in cancer chemotherapy [66], but few studies have depicted the development of dendrimers as active drugs per se, which represents a new strategy for developing new drugs [64].

In this original review, we discuss advanced dendrimer designs as promising candidates against COVID-19. We analyze the opportunity to develop biocompatible dendrimers as drugs themselves or as nanocarriers against COVID-19 infection. From the standpoint of COVID-19, we suggest developing dendrimers as shields against COVID-19 infection based on their capacity to be incorporated in several environments outside or within the various prophylactic measures developed to curtail the COVID-19 pandemic.

## 2. General Aspects of Applications of Nanotechnology in COVID-19 Treatment: A Concise Overview

In the nanotechnology approach nanoparticles (NPs) and viruses act at the same scale, which makes this strategy very powerful for treating COVID-19, including the development of nanomaterial-based vaccines. Several diverse approaches against COVID-19 using nanotechnology have been analyzed in several tutorial reviews and have opened many technology-based opportunities [67,68]. A very interesting example is in the theranostic nanoparticles field. In order to improve anti-COVID-19 treatment efficacy, intravenous injection or inhalation of potent corticosteroid drugs, such as dexamethasone, under PEGylated liposomal nano-formulation has been developed [69]. The objective is to treat infections caused by SARS-CoV-2 by the pulmonary delivery in alveolar macrophages of dexamethasone liposomes which hyper-activates the immune cells based on its anti-edema and anti-fibrotic mechanism and attenuates the production of proinflammatory cytokines. A second aspect concerns the intranasal delivery therapy. Delivery of bioactive entities including NPs to the lungs through the nasal cavity represents an interesting safe and non-invasive administration way to effectively treat viral infection due to the cavity’s abundant capillary plexus and large surface area [70]. Given that infection with SARS-CoV-2 starts on the mucosal surface of the nasal cavity as well the eye, mucosal therapy appears to be one of the most important strategies for treating such infectious diseases. The major characteristics of NPs targeting the nasal cavity with safety and efficacy are surface charge, size and shape [71]. An interesting analysis has been highlighted by Marasini and Kaminska regarding characteristics of liposome-based vaccines for their delivery through the nasal route. Epaxal^®^ and Inflexal^®^ V are already available to protect against hepatitis A and influenza, respectively [71]. The delivery of drugs and NPs including liposomes to the pulmonary respiratory system via inhalators (inhalation aerosols) represents a powerful technique to tackle COVID-19 [72]. Beyond therapeutics to fight COVID-19, personal protective equipment, such as masks, gloves and gowns, as well self-disinfecting surface technologies, have attracted considerable attention. Thus, to inactivate SARS-CoV-2 using external environmental nanotechnology, several techniques have been studied, such as (non-exhaustive list): (1) development of air filter fibers such as TiO2-coated filters [73] to capture particles (300 nm diameter size range). Development of face masks which not only can capture the aerosol droplets but can immobilize and kill SARS-CoV-2 (and other virus and bacteria). Nanographenes [74], gold, silver [75], copper [76] NPs and transition metal dichalcogenides [68], for instance under nanoflower forms, have attracted enormous attention and may be a suitable candidate for the inactivation of SARS-CoV-2. Interestingly, the development of antiviral agents shielding the viral surface and consequently preventing the viral entry into the host cytoplasm has attracted attention in antiviral research. An interesting example is the design of bovine serum albumin (BSA)-coated tellurium nanostars, preventing the cell entry of arterivirus (e.g., porcine reproductive and respiratory syndrome virus) and coronavirus (e.g., porcine epidemic diarrhea virus) [77]. Another example is the synthesis by Dey et al. of nontoxic flexible nanogels based on dendritic polyglycerol sulfate exhibiting an antiviral activity against herpes simplex virus type 1 (HSV-1) by blocking virus attachment to cell membranes [78].

Importantly, to overcome the catastrophic COVID-19 pandemic, rapid and successful development of several suitable and effective preventive vaccines has been prioritized. The four main advanced nanoplatform vaccines [67,79,80], including viral vector vaccines, virus vaccines, protein-based vaccines, nucleic acid vaccines, represent a fantastic step forward to dreadful pandemic [81]. These antiviral therapeutics use lipid-NPs [82] to package strands of mRNA, encoding SARS-CoV-2 spike protein, in engineered nanoliposomes: mRNA-1273 Moderna/NIAID and BNT162 BioNTech/Pfizer, as well as the DNA plasmid vaccine with electroporation, INO-4800, from Inovio Pharmaceuticals/CEPI/Korean Institute [81]. One advantage to the use of lipid nanoparticles, such as liposomes, in this new vaccine race (e.g., Moderna/NIAID and BNT162 BioNTech/Pfizer) is that their size is similar to that of the virus, thus enabling them to come in close proximity to attack. In addition to liposomes, other nano-formulations have been developed, such as nanocrystals, emulsions, micelles, solid lipid, linear or hyperbranched polymeric nanoparticles and dendrimers [83]. An analysis of the evolutionary fate of SARS-CoV-2 in the post-vaccination phase has been advocated by Fernandez [84]. Recently, Miao et al. emphasized a very interesting overview analysis of mRNA vaccines, including recent progress and existing challenges which have become a promising platform for cancer immunotherapy [85].

## 3. Dendrimers Used in COVID-19 Treatment: Few Clinical Examples but Limitless Possibilities

### 3.1. Dendrimers as Nanocarriers or Nanodrugs: A Concise Overview

In the dendrimer space field [86], an important direction is the development of dendrimers as drugs themselves, i.e., as the active component. In contrast to the use of dendrimers as nanocarriers for the delivery of active therapeutic agents (which is the majority of published studies), few teams are currently working on the development of dendrimers to be used as antiviral, antibacterial, antimicrobial, anticancer, antioxidant, antiviral, antitubercular and anti-inflammatory agents [64]. Figure 2 shows a global view of the dendrimers used in therapeutic domains, including conjugated or complexed SiRNAs as nanocarriers and as active drugs per se (see ref. [64] for data). Clearly, this Figure illustrates the limitless possibilities of using dendrimers as nanodrugs and nanocarriers in multiple therapeutic realms using these two strategies.

### 3.2. Dendrimer Nanoplatforms in Anti-Viral Therapeutic Domain at a Glance

Very recently, as pinpointed by Filipczak et al., much attention is now being paid to research into the use of dendrimer nanoplatforms as antiviral agents against several types of virus including: (1) human immunodeficiency virus (HIV); (2) coronaviruses; (3) Ebola virus; (4) influenza virus; (5) herpes simplex virus; (6) hepatitis C; (7) HPV (cervical cancer); (8) FMDV (foot-and-mouth disease virus); and (9) RVS (respiratory syncytial virus) [87]. In the same direction, in 2012, Mintzer and Grinstaff et al. highlighted the role of dendrimer multivalency to combat emerging and reemerging infectious disease [88]. Recently, Ortega et al. advocated for the use of dendrimers and dendritic materials against several viral infections including Enterovirus A71, Ebola virus, Zika and Dengue viruses [89].

Within non-exhaustive reported studies about the use of dendrimer platforms for potentially treating COVID-19, several papers can be highlighted. As presented by Itani et al., dendrimers potentially make strong interactions with viruses and should prevent host cell infections against COVID-19. This outstanding review analyzed the promising role of NPs such as theranostic dendrimers as effective carriers for therapeutics or immune modulators to help in fighting against COVID-19 [90]. Interestingly, similarly to SARS-CoV-2, Kandeel and colleagues studied the antiviral activity against Middle East respiratory syndrome coronavirus (MERS-CoV) with a series of 16 diverse PAMAM dendrimers [91]. In these very interesting studies, three types of polyanionic dendrimers bearing sodium carboxylate terminal groups (generations 1.5, 2.5, 3.5 and 4.5), amido-ethanol hydroxyl groups (generations 2, 3, 4 and 5), succinamic acid groups (generations 2, 3, 4 and 5) and polycationic dendrimers containing primary amine groups (generations 2, 3, 4 and 5) were designed and used in a MERS-CoV plaque inhibition assay. G(1.5)-CO2Na PAMAM dendrimer showed the most potent inhibition of MERS-CoV plaque formation (40.5% inhibition), followed by G(5)-succinic acid-PAMAM dendrimer (39.7% inhibition), bearing 16 and 128 terminal groups, respectively. The cationic dendrimers were cytotoxic against Vero cells. The authors suggested to use these dendrimers as nanocarriers by incorporating antiviral agents.

Recently, Wagner and colleagues described the preparation of amphiphilic polyphenylene dendron conjugates bearing on the surface of alternative hydrophilic (sulfonic acid) and lipophilic (propyl) groups. This surface construction serves as a biorecognition unit that bind to the surface of adenovirus 5 (Ad5), and provides high gene transduction efficiencies and high-protein binding capacity [92]. The pattern of the amphiphilic polyphenylene dendron originated from desymmetrization of amphiphilic polyphenylene dendrimers.

Using molecular dynamic simulation studies, Han and Král proposed that the binding affinity of peptide inhibitors extracted from ACE2 providing high SARS-CoV-2 blockage will be enhanced when multivalent binding of multiple peptides is attached to surfaces of nanoparticles, dendrimers and clusters [93]. In addition, these inhibitors could be used as inhaled therapeutics, preventing the virus activation in lungs.

### 3.3. Dendrimers in Clinical Phase to Tackle COVID-19

Despite the large amount of research performed over the past decades, few dendrimers have crossed the milestone of entering a clinic. Starpharma Holdings Ltd. (Melbourne, Australia) marketed the first-in-class generation three L-lysine dendrimer, VivaGel^®^ (SPL7013) [94,95], for use in antiviral and antibacterial applications, such as treatment and prevention of bacterial vaginosis and as an antiviral agent [96]. As shown in Figure 3, VivaGel^®^ bears 32 sodium 1-(carboxymethoxy) naphthalene-3,6-disulfonate surface groups. Very recently, Starpharma announced the development of VivaGel^®^ as a potent antiviral agent against the respiratory syncytial virus (RSV), for use both before and after exposure to the virus, using nasal spray technology (VIRALEZETM). Expanding studies examining additional viruses, such as the SARS-CoV-2 and influenza respiratory viruses, are ongoing [96]. Initial results showed that VivaGel^®^ inhibits replication of SARS-CoV-2 in Vero E6 and Calu-3 cells by reducing virus-induced cytopathic effects as well infectious virus release with an EC50 of approximately 0.002 mg/mL and a selectivity index of 2197. In Vero E6 and Calu-3 cells, VivaGel^®^ inactivated SARS-CoV-2 infectivity by > 99.9% (1 min of exposure). Additionally, in a primary human airway epithelial cell line, VivaGel^®^ inhibited infection via inhibition of virus-host cell interactions. Studies regarding the binding of VivaGel^®^ to SARS-CoV-2 spike protein, thus blocking the interaction of the virus with the host human membrane ACE2 protein (the first event of infection), are ongoing, as are investigations into its use as an antiviral agent against COVID-19 via nasal or inhalational administration [97]. One of the important advantages of dendrimers is that they can reach the systemic circulation via multiple administration routes, such as oral, intravenous, transdermal, topical ocular [98] (Figure 4) and nasal routes [99]. In an excellent review, Kuzmova and Minko highlighted the strong advantages of nanotechnology approaches for inhalation treatment of lung diseases [100]. This point is fully in agreement with Gatti and De Ponti regarding the development of repurposed anti-COVID-19 disease treatments using a new formulation type such aerosol administration that overcome the PK issues, such as for chloroquine, hydroxychloroquine, remdesivir and heparine [101].

Interestingly, Sahu et al. analyzed the possibilities of using biocompatible polymers as preventive measures against the ocular transmission of COVID-19 in healthcare workers, since goggles and face shields are unable to offer complete protection. The main nanoparticles highlighted and discussed in the article are chitosan, heparin, hyaluronic acid and the use of lower generation dendrimers in the prevention of viral interactions (HSV-1, HSV-II and HIV) with healthy cells [102].

Similarly, Orpheris Inc. (Redwood City, CA, USA) [103] is testing a therapeutic involving N-acetyl-cysteine coupled to the inactive dendrimer OP-101 (chemical structure not disclosed) in patients with severe COVID-19. In a randomized, double-blind, placebo-controlled Phase II clinical study (NCT04458298), OP-101 was shown to reduce COVID-19-related inflammatory cytokine storms through analysis of pro-inflammatory biomarker levels in the blood [104]. The previous therapeutic indication of OP-101 was for the treatment of childhood cerebral adrenoleukodystrophy [105].

## 4. Dendrimer Platforms for Virus Detection

Interestingly, nanotechnology based on dendrimers has also recently been advocated for in virus detection outside of patients by Farzin et al. Modification of an HT18C6(Ag) electrode with chitosan and G3 PAMAM dendrimer-coated silicon quantum dots (SiQDs@PAMAM) enabled detection of the SARS-CoV-2 virus using voltametric determination of its RNA-dependent RNA polymerase sequence [106].

One nanotechnological strategy worth noting is the development of novel SARS-CoV-2 testing and diagnosis/detection kits using dendrimers, for instance the development of sensitivity enhancement of a surface plasmon resonance sensor (SPR) [107,108].

## 5. Conclusions and Perspectives

The simultaneous development of nanotechnology and nanomaterials for use in the diagnostics and therapy of dangerous viral infections, especially COVID-19 caused by SARS-CoV-2, represents a multidisciplinary perspective encompassing diverse fields of research. Nanoparticles (e.g., gold nanoparticles, quantum dots, graphene oxide, zinc oxide, organic nanoparticles and liposomes) have been developed as novel antiviral therapeutics, for instance in the treatment of human coronaviruses [109]. To stop the spread of COVID-19, a new class of engineered and nanoarchitectured mRNA vaccines have been developed that encode the SARS-CoV-2 spike protein encapsulated within liposomes to protect it from degradation and shuttle it into cells; several of these vaccines have been marketed. Concomitantly, a strategy for repurposing existing approved small molecular drugs has been developed, with clinical trials being conducted on numerous existing antiviral drugs, such as remdesivir, hydroxychloroquine, favipiravir, pirfenidone, baricitinib, camostat, lopinavir/ritonavir [10,11] and paritaprevir [110] in search of an effective treatment for COVID-19. Additionally, several antiviral candidates are in clinical trials, such as the SARS-CoV2-3CL protease inhibitor PF-07321332 from Pfizer, as well as IL-6 inhibitors, including tocilizumab, to fight the cytokine storm associated with COVID-19 pneumonia. Virtual screening and repurposing, for instance with FDA-approved drugs against SARS-CoV-2, have also been described [111].

In the nanomedicine domain, the development of engineered, highly ordered, branched polymeric macromolecule dendrimers as active drugs themselves has been less investigated than the use of dendrimers as nanocarriers. To date, very few studies have been performed using dendrimers as first-in-class active drugs for the treatment of COVID-19 caused by SARS-CoV-2; the main actors in this field are Starpharma Holdings Ltd. (Melbourne, Australia, vide supra), which is developing a nasal spray form of VivaGel^®^ that uses VIRALEZETM spray technology, and Orpheris Inc. (Redwood City, CA, USA, vide supra), which is clinically developing the dendrimer OP-101. Interestingly, the Mediphage company (Waterloo School of Pharmacy and Professor Roderick Slavcev, Canada) is developing a nasal spray vaccine based on DNA technology using the same administration pathway [112]. Generally speaking, we fully support the development of encapsulated anti-COVID-19 treatments such as chloroquine, hydroxychloroquine and heparin with biocompatible dendrimers via aerosol administration to increase their respective PK/PD profile [70]. Currently, more than a dozen teams are working on COVID-19 vaccines that can be squired/sprayed into the nose triggering the mucosal immune system, producing specific antibodies in the nose and stopping SARS-CoV-2 infection at its point of entry [113]. We strongly recommend the development of active dendrimers against COVID-19, triggering mucosal and systemic immunity.

In parallel to ramping up the production of robotics such as ventilators [114], we fully agree with Weiss et al. [67] that prophylactic nanotechnology tools should be developed for inactivating SARS-CoV-2 in the forms of self-disinfecting surfaces, for instance using copper in place of stainless steel as an antimicrobial surface, and the widespread use of protective face masks with materials that immobilize and kill the virus, such as nanographene derivatives. In addition to the development of biocompatible dendrimers as delivery vehicles, dendrimers themselves have a place in the platform of therapeutics and are a promising candidate against SARS-Cov-2. Dendrimers can encapsulate or be conjugated to antiviral drugs, such as those against SARS-CoV-2 (vide supra). Alternatively, dendrimers can be conjugated to adenosine while also encapsulating -tocopherol to tackle the uncontrolled, virally induced hyperinflammation associated with COVID-19, as proposed by P. Couvreur et al., who developed squalene-based multidrug nanoparticles [115].

As highlighted and analyzed in Chakravarty and Vora’s tutorial paper [116], different types of nanomaterials can serve as delivery vehicles for known antiviral drugs, including lipid-based, polymer-based, lipid-polymer hybrid-based, carbon-based and inorganic metal-based nanomaterials, in addition to newer promising anti-viral treatment approaches encompassing nanotraps, nanorobots, nanobubbles, nanofibers, nanodiamonds and nanovaccines. Additionally, nanoplatforms entirely composed of dendrimers should be investigated as promising candidates in the fight against viral infections such as SARS-CoV-2; this includes the development of (glyco) dendrimersome nanoplatforms as virus-mimic nanoparticles [117]. Dendrimers also have a rightful place in these platform materials due to their capability to encapsulate metal nanoparticles, encompassing copper oxide in nanofiber-based nanofilters [118].

In *The Plague* (1943), Albert Camus warns humans to never forget the lessons of hardship with this statement: “To state quite simply what we learn in time of pestilence: that there are more things to admire in men than to despise.”

## Figures and Tables

**Figure 1 pharmaceutics-13-01513-f001:**
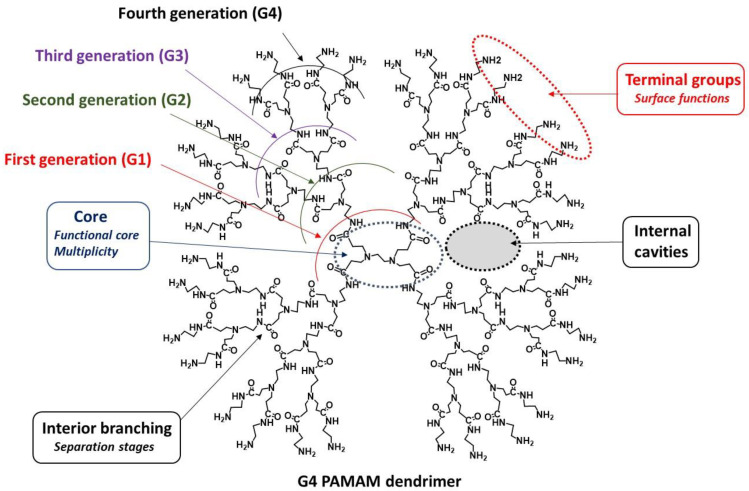
Schematic representation of 2D G4 PAMAM dendrimers and structural elements as an example.

**Figure 2 pharmaceutics-13-01513-f002:**
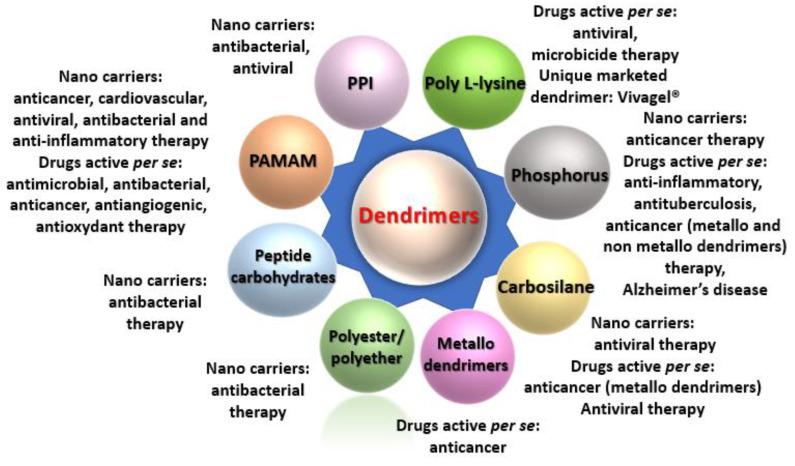
Global view of the dendrimers used in therapeutic domains, including conjugated or complexed SiRNAs as nanocarriers and as drugs. See ref. [64] for corresponding references.

**Figure 3 pharmaceutics-13-01513-f003:**
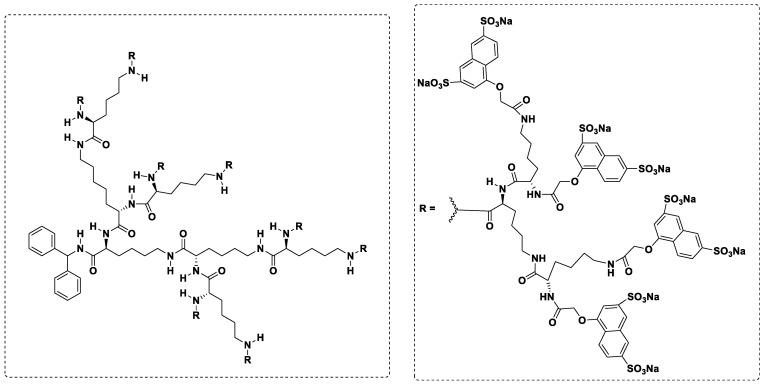
2-D chemical structure of VivaGel^®^.

**Figure 4 pharmaceutics-13-01513-f004:**
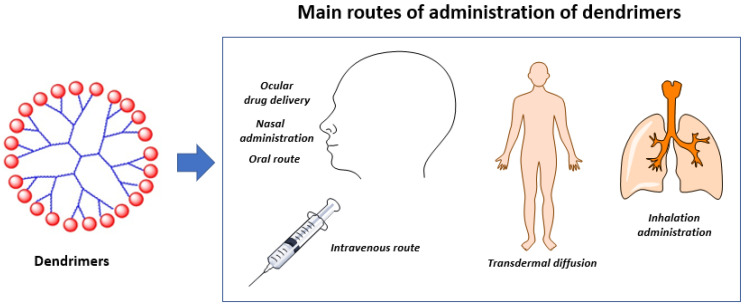
Schematic description of the main routes of administration of dendrimers.

## Data Availability

Not applicable.

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
