# Peer review of "Functionalized Dendrimer Platforms as a New Forefront Arsenal Targeting SARS-CoV-2: An Opportunity"

_pharmaceutics, 2021, doi:10.3390/pharmaceutics13091513_

Round 1

Reviewer 1 Report

This minireview entitled "Functionalized dendrimer platforms as a new forefront arsenal targeting SARS-CoV-2: An opportunity" provides an opportunity of biocompatible dendrimer to prevent the transmission of SARS-CoV-2 transmission and hence to treat COVID-19. The topic of this review is interesting and within the scope of the journal. The manuscript is nicely written and acceptable for publication after the modifications suggested below: 
The each word in the title of the manuscript should be capitalized according to journal style.

Abstract: Authors are advised to define each abbreviated term in abstract in its first appearance. All abbreviations should also be defined in main text in their first appearance.

Lines 30-48: The highlights and contents are not required for this journal. Kindly remove them.

The citation of literature in text should not be in superscript. Authors have prepared the manuscript using journal template but they did not follow the formatting and style of the template. Authors are strongly advised to follow the journal style for the preparation of this manuscript.

Figure 2: Kindly increase the size of the letters.

Figure 3: Kindly remove the colors from functional groups presented in Figure 3.

The overview of nanocarriers/dendrimers in COVID-19 treatment should be summarized in tabular form for quick understanding.

References: Kindly follow the journal style. Include complete pagination or article number in case of electronic only journals.

Various sections like author contributions, funding, institutional review board statement, informed consent statement, and data availability statement are missing. Kindly include them as these sections are required for the journal.

Author Response

Reviewer 1

This minireview entitled "Functionalized dendrimer platforms as a new forefront arsenal targeting SARS-CoV-2: An opportunity" provides an opportunity of biocompatible dendrimer to prevent the transmission of SARS-CoV-2 transmission and hence to treat COVID-19. The topic of this review is interesting and within the scope of the journal. The manuscript is nicely written and acceptable for publication after the modifications suggested below: 
The each word in the title of the manuscript should be capitalized according to journal style.

Abstract: Authors are advised to define each abbreviated term in abstract in its first appearance. All abbreviations should also be defined in main text in their first appearance.

All the abbreviation were defined

Lines 30-48: The highlights and contents are not required for this journal. Kindly remove them. We removed highlights and contents

The citation of literature in text should not be in superscript. Authors have prepared the manuscript using journal template but they did not follow the formatting and style of the template. Authors are strongly advised to follow the journal style for the preparation of this manuscript. Done in all the document which is now in right Pharmaceutics format

Figure 2: Kindly increase the size of the letters. Done

Figure 3: Kindly remove the colors from functional groups presented in Figure 3. Done

The overview of nanocarriers/dendrimers in COVID-19 treatment should be summarized in tabular form for quick understanding. As indicated in the text only one dendrimer with known chemical structure was developed by Starpharma (VivaGel®) against SARS-CoV-2

References: Kindly follow the journal style. Include complete pagination or article number in case of electronic only journals. Done All the document which is now in right Pharmaceutics format

Various sections like author contributions, funding, institutional review board statement, informed consent statement, and data availability statement are missing. Kindly include them as these sections are required for the journal. We added in the document author contributions and funding

Reviewer 2 Report

Pharmaceutics-MS# 1376982

This manuscript provides a very important and timely focus on the critical role that dendrimers have played in the pandemic and more specifically in the treatment of SARS-CoV-2.

The article is well written, well-illustrated and should be expected to attract high interest with the Pharmaceutics readership. Unfortunately, the introduction (i.e., Page 3, lines 134-147 and Page 4, lines 148-156) suffers from conspicuous omissions and poor referencing to key papers and early pioneers in the area.

As such, I recommend publication only after rewriting the introductory section mentioned above. This revision should include citations that will provide a more balanced historical picture of dendrimer conception and development.  A few suggestions may be as below:

  1. The authors profile PAMAM dendrimers in Figure 1, however, they omit certain key references that have allowed them to make the important structural claims described in this illustration.

  1. Significant design features contributing to the important role of dendrimers in nanomedicine should be mentioned [R. Kannan, et al., Intern. Med.,(2014), 276, 579-617].

  1. The critical role of dendrimer branch cell symmetry that allows PAMAM dendrimers but not poly(lysine) dendrimers to encapsulate important active pharmaceutical ingredients [Biomolecules, (2020), 10, 642, 1-59].

Author Response

Reviewer 2

This manuscript provides a very important and timely focus on the critical role that dendrimers have played in the pandemic and more specifically in the treatment of SARS-CoV-2.

The article is well written, well-illustrated and should be expected to attract high interest with the Pharmaceutics readership. Unfortunately, the introduction (i.e., Page 3, lines 134-147 and Page 4, lines 148-156) suffers from conspicuous omissions and poor referencing to key papers and early pioneers in the area. We fully agree with the reviewer. We introduced a new following chapter in this direction with corresponding references (1-4)

‘In an rapid historical way, in late December 2019, several health facilities in Wuhan located in Hubei province of China, reported groups of patients with pneumonia of unknown cause but similarly to patients with SARS and MERS symptoms including fever, cough and chest discomfort, and in severe cases dyspnea and bilateral lung infiltration.[1,2] According to hospital reports, most cases were epidemiologically linked to Huanan Seafood Wholesale Market. This market sells not only seafood but also live animals, including poultry and wildlife.[3] On December 31, the World Health Organization (WHO) was informed of an outbreak of pneumonia of unidentified cause in the Wuhan region following a public information.[4]’

As such, I recommend publication only after rewriting the introductory section mentioned above. This revision should include citations that will provide a more balanced historical picture of dendrimer conception and development.  A few suggestions may be as below:

In this direction, we added new references and a new following chapter in the Introduction

‘Dendrimers are synthetic homogeneous nano-sized symmetric macromolecules with nearly monodisperse structure. The design of dendritic macromolecules is a relatively new field in nanomedicine developed by Tomalia et al.[49] For instance, in drug delivery[50], pioneered in 1978 by Vogtle and colleagues.[51] Then, Tomalia et al. named this new class of versatile NPs “dendrimers”, formed from the two Greek words “dendros”, meaning “tree” or “branch”, and “meros”, meaning “part”.[52]’

  1. The authors profile PAMAM dendrimers in Figure 1, however, they omit certain key references that have allowed them to make the important structural claims described in this illustration. The Figure 1 is only a schematic representation of 2D G4 PAMAM dendrimers and structural elements as an example. The aim is only to present the different element of ‘classical’ dendrimers. We added new references from D. A. Tomalia and L. Peng (ref. 53 and 54) about the characteristic, synthesis and biomedical applications of PAMAM dendrimer which is the most used and developped
  1. Significant design features contributing to the important role of dendrimers in nanomedicine should be mentioned [R. Kannan, et al., Intern. Med.,(2014), 276, 579-617]Done, ref 49

  1. The critical role of dendrimer branch cell symmetry that allows PAMAM dendrimers but not poly(lysine) dendrimers to encapsulate important active pharmaceutical ingredients [Biomolecules, (2020), 10, 642, 1-59]. Done, ref 50

Reviewer 3 Report

This is an interesting article that provides an overview of how hyperbranched and monodisperse macromolecules could be repurposed as drugs in the current COVID-19 pandemic scenario. It is well written, and first provides an analysis of the contributions made by nanotechnology to address the current needs to control COVID-19 viral spread. It is subsequently followed by the unique opportunities that dendrimers offer as drugs themselves, and highlights limitations in the growth of this field. It is written by pioneers in dendrimer science, and does provide an authoritative analysis of the “dendrimer space” in its potential as therapeutical platform. I would recommend publication of this manuscript. It would have been nice to see the applications of nanotechnology, together with limited use of dendrimers as drugs tabulated, which will make it easier for the reader to comprehend and analyse what is available. It may also be useful in building some structure-property relationships.

Author Response

Reviewer 3

This is an interesting article that provides an overview of how hyperbranched and monodisperse macromolecules could be repurposed as drugs in the current COVID-19 pandemic scenario. It is well written, and first provides an analysis of the contributions made by nanotechnology to address the current needs to control COVID-19 viral spread. It is subsequently followed by the unique opportunities that dendrimers offer as drugs themselves, and highlights limitations in the growth of this field. It is written by pioneers in dendrimer science, and does provide an authoritative analysis of the “dendrimer space” in its potential as therapeutical platform. I would recommend publication of this manuscript. It would have been nice to see the applications of nanotechnology, together with limited use of dendrimers as drugs tabulated, which will make it easier for the reader to comprehend and analyse what is available. It may also be useful in building some structure-property relationships.

It is not in the scope of this review to perform some structure-property relationships of dendrimers in general

Round 2

Reviewer 1 Report

The authors have addressed the previous concerns successfully. It can be accepted in its present form.